# What Stirs Consumers to Purchase Carbon-Friendly Food? Investigating the Motivational and Emotional Aspects in Three Studies

**Elfriede Penz [1,\*] and Eva Hofmann [2]**

1    Institute for International Marketing Management, Vienna University of Economics and Business, 1020 Vienna, Austria
2    Institute of Psychology, University of Graz, 8010 Graz, Austria; eva.hofmann@uni-graz.at
*    Correspondence: elfriede.penz@wu.ac.at; Tel.: +43-1-31336-5102

**Abstract:** As part of diminishing climate change, food consumption needs to be addressed to reduce greenhouse gases. In order to change food consumption habits to carbon-friendly eating patterns, consumers may be targeted by information campaigns and legal regulation. The current paper studies consumers' diets and food purchase behavior. In particular, it aims to understand consumers' motivational and emotional aspects that influence their behavior. Study 1, an interview study, aims to understand the development of and motivations for climate-friendly nutrition. Identifying eco-friendly motives also revealed that emotions seem to play an important role in nutrition and the purchase of climate-friendly products. Study 2 aims at identifying consumers' positive and negative emotions when it comes to consuming carbon-friendly food. Again, qualitative interviews revealed a variety of positive and negative emotions. Study 3 quantitatively tested the theory of planned behavior, including positive and negative emotions and predicted carbon-friendly food purchases. The results show that attitudes, perceived behavioral control and positive emotions predict carbon-friendly food purchases. Derived from these findings, recommendations for information campaigns and legislation to foster carbon-friendly food purchases are presented.

**Keywords:** carbon-friendly food; theory of planned behavior; emotions





## 1. Introduction

For decades, climate change has been a pressing problem of humanity, becoming increasingly challenging and central from year to year. Food consumption is responsible for up to 30 percent of greenhouse emissions in Western countries [1]. Thus, not only regulators, but also consumers, are asking how greenhouse gas emissions (especially carbon dioxide ($CO_2$)) from food consumption can be reduced so that climate change can decelerate. One approach is to change food consumption habits to carbon-friendly eating patterns utilizing information campaigns and legal regulation. The current research aims to build on established theories to understand antecedents and drivers of consumer behavior regarding carbon-friendly food, to make sure that powerful campaigns and legal regulation can be designed to meet the end of increasing carbon-friendly food consumption.

Such campaigns and legal regulations are particularly important as consumers are often unaware of which behaviors can be classified as carbon-friendly [2,3]. Guidebooks [4] for consumers offer a variety of behaviors on how to reduce $CO_2$ in everyday life. Concerning food consumption, consumers should focus on (a) changing the kinds of foods they consume (e.g., more plant-based diet instead of animal products), as well as consider (b) the production process (e.g., more organic food, but less processed food), (c) the method of transportation (e.g., local food, minimizing chilled or frozen food) and (d) the packaging (e.g., no or minimal packaging). Thus, consumers have several ways of reducing greenhouse gas emissions and, therefore, also $CO_2$ by changing their food consumption habits.

Introducing these behavioral changes can decrease the individual greenhouse gas and $CO_2$ emissions related to food consumption by 85 percent [4].

To create information campaigns and as a basis for regulation, businesses and legislators need to base their efforts on what antecedents and drivers determine consumers' food consumption practices. The theory of planned behavior (TPB [5]) is an adequate model to point out the relevance of different motivational aspects (attitudes, subjective norms, perceived behavioral control). While the TPB postulates that motivation (termed intention in the original publication) is strongly related to actual behavior, some research [6] indicates that one essential motivational aspect is missing in the TPB, which is necessary to explain sustainable behavior. This research gap can be addressed by investigating consumers' emotions and incorporating them into the TPB. According to Parrott [7], there are six different primary emotions: sadness, love, anger, joy, surprise and fear, that altogether can be summarized into negative and positive emotions and which impact consumers' carbon-friendly food-related behavior. Therefore, the current research aims to investigate the relationship of motivational aspects as in the TPB (attitudes, subjective norms, perceived behavioral control) and emotions with consumers' purchase behaviors regarding carbon-friendly food.

Based on the investigation of the TPB, including emotions, the research is structured as follows: first, it gives an overview of the theoretical background focusing on carbon-friendly food behavior, the TPB and emotions, whereby this chapter closes with the consolidation of the TPB and emotions in consumer behavior and presents the theoretical research model. Second, it presents three empirical studies: two explorative interview studies and one representative questionnaire study, which examine the relationship between motivations and carbon-friendly food behavior (Study 1), the relationship between emotions and carbon-friendly food behavior (Study 2) and the overall theoretical research model integrating the TPB and emotions to explain carbon-friendly food behavior (Study 3). The research is concluded with a discussion on the results of the three studies from a theoretical and practical perspective.

## 2. Theoretical Framework

Inducing consumers to purchase carbon-friendly food is certainly, on the one hand, an integral approach to reduce greenhouse gas emissions, specifically $CO_2$; on the other hand, it is a difficult endeavor that has been sought for some years now but was not easily achieved. Specifically, the determinants of carbon-friendly food purchases are not clear. However, the current research sheds light on these determinants, i.e., the factors of the theory of planned behavior (TPB: attitudes, subjective norms, perceived behavioral control [5]) and emotions [7]. Therefore, the present theoretical framework touches on carbon-friendly food purchases in general, on the determinants (TPB [5], emotions [7]) stipulating carbon-friendly food purchases, on the incorporation of emotions in the TPB and the consequential research questions.

### 2.1. Carbon-Friendly Food Purchasing

Although the majority of consumers are talking about the pressing need to reduce $CO_2$ in the atmosphere, it is not totally clear how this can be achieved with consumer choices regarding food purchases. Several aspects define whether a food item can be perceived as carbon-friendly: (a) the kind of food is crucial for $CO_2$ emissions, meaning that a plant-based diet instead of animal comestibles is less responsible for $CO_2$ emissions [8,9]. (b) Another aspect essential for $CO_2$ emissions based on food consumption is the production process of the respective food. Organic food, in general, produces less $CO_2$ in the production process, unlike processed food, such as frozen pizza or microwave dinners [9], which are responsible for much more $CO_2$ in the production process. (c) Additionally, the mode of transportation of food is very important. Food that is transported a very short distance, i.e., regional food and fresh food that needs neither chilling nor freezing, is connected to less $CO_2$ than food transported long distances in a chilled or even frozen

state [8]. (d) A final aspect of $CO_2$ pollution based on food consumption is a not-so-obvious feature, namely food packaging. The less packaging a food item is wrapped in, the fewer $CO_2$ emissions are produced. Thus, no packaging would be optimal for all food items. Nevertheless, this is not always possible because of legal hygiene standards, e.g., meat needs to be wrapped [4]. This is a rather long and diverse list of how carbon-friendly food can be defined.

Unfortunately, these different aspects of the definition do not always go hand in hand with each other. For instance, some fresh fruit might be organic and therefore carbon- friendly, but it might reach the consumers after a long-distance transport from another continent. Therefore, these fruits cannot be described as carbon friendly. Another example is fresh vegetables, which are certainly carbon-friendly in comparison to frozen vegetables, but they might be packed in several layers of plastic to make transportation more convenient. Again, such items cannot be defined as carbon-friendly food due to the excessive packaging. Under these circumstances, it is difficult for consumers to decide which food items to purchase to stop the excessive increase in $CO_2$ emissions. Nevertheless, more and more consumers are determined to change their purchase behaviors and acquire carbon-friendly foods.

Thus, if consumers are truly interested in buying carbon-friendly food, what is driving them to do so? There has been research on the drivers of purchases of carbon-friendly food. For instance, consumers' knowledge regarding the effect of $CO_2$ on the climate and the concern regarding the risk of climate change could be predictors for the willingness to spend more money on carbon-friendly food [10]. The surprising finding was that neither knowledge nor concerns significantly impact paying extra for carbon-friendly food. Thus, being aware of the problem is not sufficient to change consumer behavior. Other predictors could be different values, attitudes, social norms and perceived behavioral control. Aertsens et al. [11] collected research on the drivers of consumers who were purchasing organic food, and found that values such as security, hedonism, universalism, benevolence, stimulation, self-direction and conformity, when linked to organic food, positively impacted attitudes towards the purchase of organic food. Additionally, they showed that these attitudes, social norms and perceived behavioral control influenced the purchase and consumption of organic food. Thus, personal factors such as values, attitudes and perceived behavioral control definitely affect organic food purchase behavior. Therefore, labeling products as carbon-friendly might stimulate specific values, attitudes and social norms that stimulate consumers to buy carbon-friendly food. Studies [12,13] found a clear connection between labels for carbon-friendly food and the willingness to purchase such food items. Thus, a variety of different drivers to purchase carbon-friendly food has already been researched. Whereas some drivers are effective (personal aspects such as values, attitudes and social norms; labels), others do not impact purchase behavior (knowledge on climate change, concern regarding climate change).

Although some drivers for carbon-friendly food purchases have been identified by now, thus far the psychological process has not been fully detected. While values, attitudes, social norms and perceived behavioral control are psychological factors that have an impact, other psychological determinates of carbon-friendly food purchase are missing. As consumer research [14] shows, consumers' emotions have a significant effect on purchase behavior. Thus, why should consumers' emotions not also determine the purchase of carbon-friendly food? A preliminary empirical study indicates that emotions play a role in purchasing organic food [15]. With the current studies, we incorporate emotions into the purchase decision process and investigate how emotions, attitudes, subjective norms and perceived behavioral control impact the purchase of carbon-friendly food.

### 2.2. Theory of Planned Behavior

The theory of planned behavior (TPB [5]) is often applied in consumer behavior research that bases purchase decisions on three main determinants: attitudes, subjective norms and perceived behavioral control towards the purchase. These determinates again

result in the intention to purchase, which finally ends in the actual purchase. While this theory was originally developed to explain the influence of significant others as a social-psychological theory, it was adopted by consumer research and used to explain purchase behavior of different kinds [5,16–19].

Describing the three determinants, the theory of planned behavior [5] postulates that attitudes towards a certain behavior—in our case, the purchase of carbon-friendly food—are the evaluations of the behavior as positive or negative [20]. Thus, some consumers believe that buying carbon-friendly food is a vital behavior that needs to be undertaken to slow down climate change. In contrast, other consumers think that buying carbon-friendly food is a waste of money because climate change is not man-made and, for that reason, cannot be detained by humans. Thus, while the first consumers positively evaluate the purchase of carbon-friendly food, the second ones think of the same behavior negatively. Regarding subjective norms, a social aspect is central: subjective norms specify what significant others expect the decision-maker to do and to what extent the decision-maker wants to follow the others' claims. In the case of carbon-friendly food purchases, the person who decides to purchase carbon-friendly food can be influenced by other important persons, who indicate that purchasing carbon-friendly food is important. Additionally, subjective norms also include the consideration of the decision-maker, whether to follow the recommendations of significant others or not. The third determinant, i.e., perceived behavioral control, is defined as the perceived effort deciders have to put into the undertaking of the behavior. Regarding carbon-friendly food purchases, consumers consider how easy or difficult it is to purchase carbon-friendly food. For instance, aspects are summarized, including whether consumers have the necessary knowledge to recognize carbon-friendly food under all the possible food options, or if there is a store nearby to buy carbon-friendly food from. Thus, all three determinants of the theory of planned behavior are excellent drivers of carbon-friendly food purchases.

These three determinants (attitudes, subjective norms, perceived behavioral control) influence the intention to undertake the respective behavior [20]. The intention to undertake a certain behavior is seen as a person's motivation to undertake the behavior. With carbon-friendly food purchases, the intention, i.e., motivation, is the driver to actually buy carbon-friendly food, go to the shop and actively look for food that is low in $CO_2$ production. This motivation or intention is then directly linked to behavior. Although there is a direct link, the connection is not 100 percent. There might be a strong motivation to undertake the respective behavior, but the behavior is still not performed. In this case, perceived behavioral control could impact the behavior. If there is no possibility perceived as to how to undertake the behavior, the behavior is not performed despite the strong motivation to do so. For carbon-friendly food purchases, this specifically means that if consumers perceive that there is no shop nearby that sells carbon-friendly food, they will not buy such items, even if highly motivated to do so. In summary, the theory of planned behavior is an excellent theory to predict the purchase of carbon-friendly food [16], nevertheless one important aspect seems to be missing in this theory, i.e., emotions [21], to precisely predict carbon-friendly food purchases.

### 2.3. Emotions

Emotions are defined as "a mental state of readiness that arises from cognitive appraisals of events or thoughts; is accompanied by physiological processes; is often expressed physically (e.g., in gestures, posture, facial features); and may result in specific actions to affirm or cope with the emotion, depending on its nature and meaning for the person having it" [22] (p. 184). Therefore, emotions are an important psychological determinant of decision-making, thus also in deciding to purchase carbon-friendly food. Nevertheless, there are several taxonomies of emotions [23,24], so in the current manuscript we focus on the list of emotions in social psychology [7].

This list of emotions differentiates between primary, secondary and tertiary emotions [7], whereby a number of emotions on the tertiary level (e.g., arousal, desire, lust,

passion, infatuation) comprises one emotion on the secondary level (e.g., lust) and similarly, a number of emotions on the secondary level (e.g., affection, lust, longing) comprises one emotion on the primary level (e.g., love). For all emotions on the primary and secondary level [7], see Table 1. In the current study, we focus on these emotions, specifically on emotions on the primary and secondary levels, following other research in consumer behavior that highlights this systematization [25,26].

**Table 1.** Primary and secondary emotions [7].

| Negative Emotions | | Positive Emotions | |
|---|---|---|---|
| **Primary Emotion** | **Secondary Emotion** | **Primary Emotion** | **Secondary Emotion** |
| Anger | Irritation | Love | Affection |
| | Exasperation | | Lust |
| | Rage | | Longing |
| | Disgust | Joy | Cheerfulness |
| | Envy | | Zest |
| | Torment | | Contentment |
| Sadness | Suffering | | Pride |
| | Sadness | | Optimism |
| | Disappointment | | Enthrallment |
| | Shame | | Relief |
| | Neglect | Surprise | Surprise |
| | Sympathy | | |
| Fear | Horror | | |
| | Nervousness | | |

Although the various taxonomies differentiate between many different emotions [7,23], some taxonomies fall back on a simple differentiation between positive and negative emotions [24]. Although we value the fact of being able to differentiate between various qualities of emotions, in the current manuscript, we also summarize the six primary emotions [7] into negative (sadness, anger, fear) and positive emotions (joy, surprise, love). We consider the different characteristics of these six emotions but reduce the complexity of dealing with their manifoldness.

Earlier research has already focused on specific emotions regarding sustainable consumer behavior. For instance, guilt as a negative emotion was investigated in the context after the consumption [27], showing that the consumption of unsustainable products can stipulate guilt in consumers. Contrarily, the consumption of sustainable products can provoke the positive emotion of pride [27]. Although research on the impact of emotions on the purchase and consumption of sustainable products exists, to our knowledge, studies on the impact of manifold emotions on carbon-friendly food are scarce. Therefore, the current studies focus on the list of emotions in social psychology [7], investigating their relation to the purchase behavior of carbon-friendly food.

*2.4. Theory of Planned Behavior & Emotions*

As stated above, we build on the theory of planned behavior [TPB, 20] and adjust it with additional variables, namely negative and positive emotions [7]. As earlier research [21] has identified, although the TPB [5] is a very comprehensive social-psychological theory excellently predicting behavior, it is missing one essential aspect: the state of emotions when humans consider undertaking a behavior. A meta-analysis [28] shows that the incorporation of emotions enhances the predictive power of the TPB [5]. This is the reason why we base our theoretical model on the TPB [5] and emotions [7].

Earlier research [6,29] did incorporate emotions; specifically, the emotions of regret or fear, in the TPB [5] to explain sustainable food-related behavior, i.e., purchasing organic food or selecting an eco-friendly restaurant. However, to our knowledge, the whole range of emotions was never incorporated. Furthermore, emotions were shown to affect purchasing intentions for organic food [15] but independently from the TPB [5]. Nevertheless,

carbon-friendly food purchases incorporate more than only the purchase of organic food. For that reason, we not only incorporate a whole range of emotions [7] in the TPB [5], but also investigate the predictors of the comprehensive behaviors of carbon-friendly food purchases, i.e., plant-based diet instead of animal products, more organic food but less processed food, preferring local food over chilled or frozen food and no or minimal packaging. Based on all these considerations, we formulated the following research questions:

*Research Question 1:* What are the motivations to purchase carbon-friendly food?
*Research Question 2:* Which emotions emerge with the purchase of carbon-friendly food?
*Research Question 3:* Can the theory of planned behavior [5], including negative and positive emotions, explain the purchase of carbon-friendly food?

Figure 1 illustrates the proposed relationships that will be tested in our studies. Starting with the TPB [5] and incorporating positive and negative emotions [7], the model is supposed to explain comprehensively different kinds of carbon-friendly food purchases.

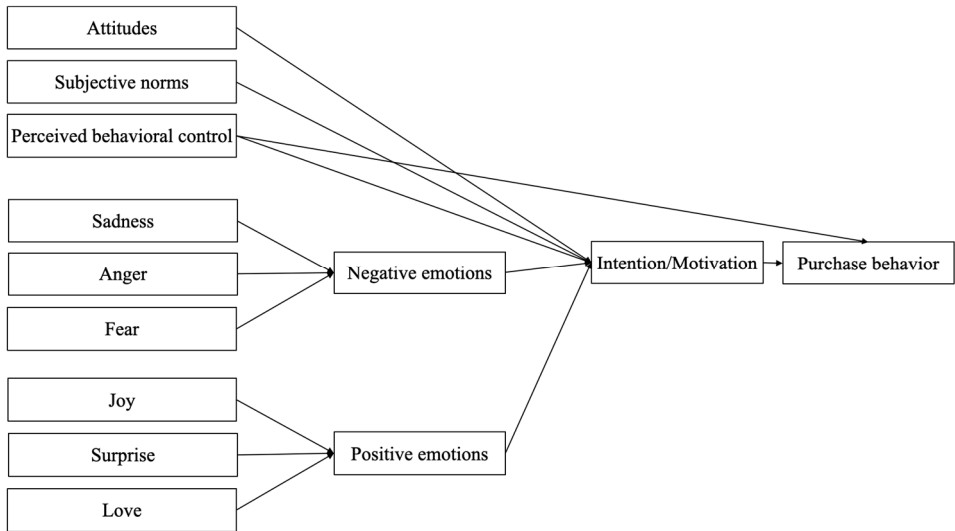

**Figure 1.** Theoretical model including emotions [7] in the TPB [5].

Answering these research questions, two explorative interview studies and one representative questionnaire study were conducted. They examine the relationship between motivations and carbon-friendly food behavior (Study 1), the relationship between emotions and carbon-friendly food behavior (Study 2) and the overall theoretical research model integrating the TPB and emotions to explain carbon-friendly food behavior (Study 3).

## 3. Study 1

### 3.1. Method

To answer *Research Question 1* (*What are the motivations to purchase carbon-friendly food?*), we searched for participants who adhere already to a carbon-friendly diet, such as vegetarian or vegan diets, and ten participants agreed. Eight out of the ten participants were female, which is acceptable since in general female respondents are more willing to participate in such studies. Their ages ranged between 19 and 53 years (M = 29.80, SD = 10.86). Six participants were students, and the others worked in various professions. Seven participants adhered to a vegetarian diet, and three adhered to a vegan diet, whereby all of these participants were vegetarians before becoming vegan (Table 2).

The interviews were conducted in German, audio-recorded and transcribed afterward. The developed interview guidelines aimed at studying motivations concerning carbon-friendly food behavior. The different parts included (1) questions about participants' vegetarian or vegan diet, (2) their motives for changes in their diet as well as (3) reactions from the social environment. In addition, aspects concerning the (4) actual purchasing

behavior, such as the place of purchase and the product itself, were discussed. In this paper, part (3) will not be considered further.

**Table 2.** Socio-demographics of participants in Study 1.

| Coded Name [1] | Sex | Age | Occupation | Diet |
|---|---|---|---|---|
| John | M | 23 | Student | Vegetarian |
| Nina | F | 23 | Student | Vegetarian |
| Lisa | F | 20 | Student | Vegetarian |
| Max | M | 24 | Student | Vegan |
| Sarah | F | 33 | Biologist | Vegetarian |
| Sandra | F | 39 | Student | Vegetarian |
| Melanie | F | 39 | Master tailor | Vegan |
| Laura | F | 22 | Student | Vegan |
| Maria | F | 22 | Student | Vegetarian |
| Eva | F | 53 | Flight attendant | Vegetarian |

[1] The coded name is fictitious and not related to the actual name of the participant; M = male, F = female.

Data were analyzed following qualitative content analysis [30] to identify relevant categories and patterns to explain the relationship between motivations and carbon- friendly food behavior.

*3.2. Results*

After a comprehensive analysis, it turns out that mainly four groups of motivations for a vegan as well as a vegetarian diet prevailed: (1) ethical concerns about animal rights, (2) personal health concerns, (3) environmental sustainability and (4) disgust towards meat. While all these motivations were present for both vegans and vegetarians, some motivations were more present for one or the other group.

Regarding (1) the ethical concerns about animal rights, respondents named animal suffering and cruel living conditions of animals in factory farming as reasons for stopping eating meat. For instance, Nina (23 yrs.) informed herself about the living conditions of animals and decided to become a vegetarian. Max (24 yrs.) explained his vegan diet by saying, "Each and every living being is valuable and eating meat is simply wrong" (all verbal quotes were translated from German into English). In a similar vein, Sarah (33 yrs.) describes her feelings as follows: "It always has been an effort to eat dead animals; the ethical aspect prevailed at my side." In sum, more than half of the participants listed animal rights as the dominant motivation for their vegan or vegetarian diet.

(2) Acute health problems or wanting to do something for their personal body health, in general, were important reasons for respondents to change their diets; particularly for those becoming vegan, personal health was a major driver. For instance, Eva (53 yrs.) stated, "I followed a vegetarian diet because I wanted to do something for my health. At that time, I was sure that it was beneficial for my health, which is why I started step by step." Some respondents realized the impact of animal protein on their health. Respondents mentioned a documentary [31] as having had a major impact on their diet change. Not only did the documentary reveal the shocking effects of Western diets on health, but it also presented a solution to the problem, i.e., a mainly plant-based diet. Laura (22 yrs.) suffered from lactose intolerance and became vegan.

(3) Environmental sustainability was mentioned in a rather broad sense. For example, Melanie (39 yrs.) states, "I care about the climate. I am a bit worried about the environment." Lisa (20 yrs.) also says, "I believe the consumption of meat can be a severe problem for our environment. Well, the intense factory farming (of meat)." More specific aspects for becoming vegan or vegetarian related to resources, for instance, as Sandra (22 yrs.) puts it, "Once you realize how much water is needed to eat one kilogram of beef compared to eating one kilogram of beans, both having similar nutritional values, then it is a difference of 10,000 or 20,000 kg of water, I guess." However, environmental sustainability did not

appear to be the sole concern for our respondents, but mentioned in conjunction with animal rights and health concerns.

Another reason behind becoming vegan or vegetarian includes (4) disgust or dislike of meat. The meat-specific unpleasant experiences of taste, smell, look or texture were mentioned. Lisa (20 yrs.) argued that initially, she thought that animal welfare influenced her but she realized that "it did not taste pleasant. I justified becoming vegetarian by saying 'I feel sorry for animals', but it was more that it didn't taste good." Emotional reactions such as dislike or disgust towards meat often represented a basis for the change to a vegetarian or vegan diet but seldom served as principal motivation.

For most of the respondents, multiple motivations were relevant. Some of these became more important over time (e.g., environmental sustainability), partly because of increased exposure to information and increased awareness. As Sandra (22 yrs.) puts it: " . . . the environmental aspect also turned up, I wasn't aware of it earlier." In this respect, it was interesting to hear that previously vegetarians changed to a vegan diet, and their transition seems to be easier and quicker because they already have some knowledge and experience in this field. For instance, Melanie (39 yrs.), now vegan, states: "I was vegetarian before. I actually tried it all my life. I mean, as a child, I was not allowed to, but I refused to eat meat most of the time."

Other aspects were also important for participants' motivation to engage in carbon-friendly food behavior. Regarding the origin and production of products, the majority of participants (seven out of ten interviewees) listed either organic or regional as important characteristics of food for their purchasing decision. Vegetarian consumers tend to purchase more carbon-friendly compared to vegans. Several vegetarians explained in detail how important the origin and the way of production are for their choice of products. This contrasts with some vegans, who mentioned that they might buy organic products sometimes or do not pay too much attention to them.

In this respect, some mention the importance of the price–performance ratio for their buying decision and, owing to that, reject buying organic apples, instead choosing regional ones. As John (23 yrs.) states: "When it comes to buying apples, which sometimes cost up to three times as much when they are organic, it is sufficient to buy Austrian apples." Moreover, all participants purchased their food mainly in supermarkets, as the convenience and price were appealing. Of course, some bought certain products in organic supermarkets, but in general, limited budgets were constraints.

Regarding substitute products, only one interviewee mentioned that she regularly purchases vegetarian substitute products (such as vegetable patties). Reasons for not adopting substitutes related to perceived unhealthiness are the high amount of food additives included in those products or tastes. Instead, alternatives were consumed; for instance, Laura (22 yrs.) states, "I hardly buy substitute products. I think, in the first year, I mainly ate only (a substitute) yogurt. It is now one year that I've tried out some . . . But I don't think that they should be a fixed part of my diet." Food additives play an important role in purchasing decisions in general. Half of the participants explicitly noted that they pay attention to additives and binders included in certain products when buying food.

Labels of vegetarian and vegan products were discussed highly controversially during the interviews. A clear and consistent marking seems to be important to clarify the ingredients of a product, which is important for vegetarian and vegan consumers. Nevertheless, the image and perception others have regarding vegetarians and vegans and their lifestyle is a critical issue. Therefore, labeling or packaging which highlights the product as vegetarian or vegan, in a too dominant, overdesigned manner is perceived as unattractive. Two vegetarian interviewees complained about the packaging and the labeling of vegetarian products. Particularly, they mentioned the green color of the products as well as the label "vegetarian." As a vegetarian, they do not want to be excluded from other consumer groups. Lisa (20 yrs.), for instance, argues, "They give you the experience of being different, but in a negative way." Another respondent finds the green packaging of vegetarian products misleading and argues that this choice of color should imply that the product is healthy

which, is often not the case for vegetarian food. Maria (22 yrs.) says, "I find packaging usually disgusting because of their green color. Vegetarian is automatically perceived as healthy, which is nonsense." Nevertheless, even if it is clear for some products, the labeling "vegan" still facilitates the purchase of food.

*3.3. Discussion*

An important reason for switching their diet is concern about animals raised for food [32]. Ethical concerns represent the major motivations for vegetarians to stop their meat consumption. Consumers state in particular animal rights and the quality of life of animals as main concerns in this regard [33,34]. We also identified personal health as the main influencer for the adoption of a vegetarian diet. Moreover, consumers expressed negative emotions, such as disgust and dislike regarding meat consumption, indicating that emotions are important drivers to pursue carbon-friendly food behavior.

Ecological sustainability is getting increasingly important for consumers who follow a vegan diet for some time [32], which was also observed in our study; in addition, the engagement of persons on plant-based nutrition tends to rise over time. Market-related factors such as retail stores, product offerings and labeling as well as pricing are taken into consideration when shopping carbon-friendly, in our case for vegan or vegetarian food products.

**4. Study 2**

*4.1. Method*

To answer *Research Question 2* (*Which emotions emerge with the purchase of carbon-friendly food?*), participants' diet was not treated as selection criterion to allow enough breadth of insights into consumers' emotions.

Seven out of the ten participants were female, which again is acceptable since in general female respondents are more willing to participate in such studies. Overall, the age range was between 23 and 60 years (M = 34.20, SD = 14.80). Half of the sample were students; the others were working in various jobs. Regarding the participants' diets, five were omnivores, eating all kinds of foods, three were flexitarians, mainly focusing on vegetables and dairy products with sometimes meat or fish, and two were vegetarians, eating vegetables and dairy products (Table 3).

**Table 3.** Socio-demographics of participants in Study 2.

| Coded Name [1] | Sex | Age | Occupation | Diet |
|---|---|---|---|---|
| Mara | F | 23 | Student | Flexitarian |
| Linda | F | 24 | Student/Part-time Job | Omnivore |
| Hannah | F | 23 | Student | Omnivore |
| Stefanie | F | 54 | Office worker | Omnivore |
| Anna | F | 52 | Architect | Omnivore |
| Jakob | M | 26 | Student/Part-time Job | Vegetarian |
| Charlotte | F | 28 | CEO of an organization | Vegetarian |
| Franz | M | 60 | Office worker | Flexitarian |
| Viktoria | F | 25 | Student/Part-time Job | Omnivore |
| Stefan | M | 27 | Technician | Flexitarian |

[1] The coded name is fictitious and not related to the actual name of the participant; M = male, F = female; Flexitarian = a mainly vegetarian diet with meat or fish sometimes, Omnivore = a diet consisting of meat, fish, dairy products and vegetables, Vegetarian = a diet consisting of vegetables and dairy products.

All interviews were conducted in German, audio-recorded and transcribed afterward. The study aimed at assessing the relationship between emotions and carbon-friendly food behavior. The interview guidelines of this study followed a structured market research method, i.e., ZMET [35]. Part of this technique was to instruct participants (about one week prior to the interview) to think about food that is produced organically or is producing less $CO_2$ in the production, transportation, consumption and disposal process, and to collect

10–15 pictures (independent of medium) that express their respective thoughts. Data were analyzed following qualitative content analysis [30] with the aim of identifying relevant categories and patterns to explain the relationship between emotions and carbon-friendly food behavior.

(1) The first step of the ZMET interview included "storytelling." Participants were asked to describe each of the pictures and their related thoughts and explain possible issues that may relate to the pictures. The closing question in step 1 was on participants' feelings when purchasing sustainable food products. (2) The second step of the technique was to let participants sort their pictures into meaningful categories. (3) Step three ("most representative picture") included the question about which picture is best associated with the topic. (4) In step four ("missing images"), it was asked whether participants were unable to find a specific picture; while in (5) step five, participants were asked to select one picture that is least associated with the topic ("opposite image"). Eventually, (6) in step six ("summary image"), the participants created a summary image (collage) of all collected pictures as part of the interview.

Concerning the "most representative image," participants selected a variety of pictures (e.g., a farmers' market with an elderly woman selling products, seasonal vegetables, a wheat field, a quotation of Mahatma Gandhi, etc.)

Regarding "missing images," nearly all participants found all the pictures they were looking for. When looking at the "opposite image," participants selected a variety of pictures (the brand "JA! Natürlich" (products and the mascot pig), sign with the word "sustainability" on it, mountain water, a ray in the sea, etc.).

### 4.2. Results

In order to analyze the relationship between emotions and carbon-friendly food behavior, the participants categorized their collected pictures. The most often used category was "own products/own garden/home-made" (f = 7). Five participants choose the category "local products/local farmers' markets/local resources." In addition, categories "seal of quality/food brands" (f = 3), "organic farming" (f = 2) and "zero-waste/packaging/model for the future" were used to group pictures. All other 23 categories were only mentioned once, e.g., "the first step to sustainability," "natural protein source" or "shopping." Overall, participants sorted their pictures into 28 different categories.

Regarding the question of which emotions were evoked, in the following negative and positive emotions will be described, following the framework of emotions by Parrott [7].

#### 4.2.1. Positive Emotions

The positive emotions, which the participants mentioned, are the primary emotion joy, with its secondary emotions optimism (joy) and pride (joy), and its tertiary emotions happiness (joy), satisfaction (joy), enthusiasm (joy) and desire (love). Additionally, other emotions were stated, such as affiliation, trust and feeling good.

When thinking of sustainable food consumption, participants named joy in connection with growing their own vegetables. For instance, Hannah (23 yrs.) explained: "It somehow shows me that you can enjoy fruits and vegetables that grow in their natural habitat, which have not artificially been produced." Stefanie (54 yrs.) feels joyful when she buys organic food products at farmers' markets. She said: "Farmers from nearby come to offer their organic products here, and it is such a joy to walk through (the market) and experience the seasonality of their products."

In one specific case, a participant felt joyful when she collected chanterelle mushrooms in the forest. Shopping in a zero-waste supermarket induced joy for Stefan (27 yrs.), who mentioned it as a great opportunity to reduce waste from packaging. In addition, Anna (52 yrs.) feels joyful when she thinks about the different forms of vegetables that are available: "It brings me unbelievable joy to have such a variety, and I can imagine that each form is related to different substances. That all these pumpkins differ in terms of colors, smells and tastes."

Many participants feel optimistic about the future's sustainable food production and consumption. They think that everyone can contribute to a fairer allocation of the world's resources. For instance, Linda (24 yrs.) is sure that every step counts; according to her, "Everyone can make a difference. No matter how small it is, for instance, instead of eating meat three times a week, eat it only once a week. I believe this is extremely valuable; if everyone does that, it sums up. Even a small amount will make a difference."

Pride was also mentioned a lot by the participants. Mara (23 yrs.) is proud that the family grows their own vegetables, "one is very proud about having themselves collected or grown it and then cooks it themselves," and Linda (24 yrs.) mentions that her family's diet is a source of pride and unites them. She says, "Not every family values these things. I think my children are very proud and are happy about this too."

Some participants feel happy and satisfied if they prepared food on their own or eat the vegetables they grow on their own. For instance, when growing her own vegetables under difficult conditions, Anna (52 yrs.) feels satisfied, as her work was successful. She says: "I was so happy that for the first time this year we succeeded in growing carrots. These carrots are a symbol of success and joy in the unexpected success."

The enthusiasm, which is felt if they consume sustainable food, can be illustrated with Franz's (60 yrs.) statement about his homegrown garlic: "I eat it now with enthusiasm, and I couldn't care less about the imported garlic."

In a similar vein, a special sustainable food product evoked desire (love) as Stefan (27 yrs.) expresses it as follows: "The bread looks really crunchy and fresh. You just want to eat it!"

Some respondents feel affiliated with their families when they think of consuming sustainable food. As Linda (see above regarding optimism) did, Stefanie (54 yrs.) also mentioned that she feels close to people who have the same interests as her. According to her, the feeling "of solidarity, of being a part of a group or community, evokes a strong feeling of unity."

Participants talked about trust when thinking of ecologically sustainable food brands, such as Mara (23 yrs.), who states, "You can't know for sure, I have no control over it, but I trust products are organic if they come from organic agriculture." Food produced in Austria is trusted more; for instance, Jakob (26 yrs.) says, "I ultimately trust products made in Austria more."

Eventually, a more general positive expression was used: to feel good. If a chicken lives a good life, Mara (23 yrs.), for instance, feels good too. For Jakob (26 yrs.), too, spending his holidays on farms makes him feel good, and he is not surprised that a supermarket's private brand uses farm images to evoke such nostalgic feelings. A variety of animals and plants is mentioned in this context and in general, as Franz (60 yrs.) states: "The most important thing is to sustain the variety of species and plants to preserve a habitat where you feel good and can live a happy and healthy life."

### 4.2.2. Negative Emotions

We identified negative emotions that are categorized by Parrott [7] as the primary emotion of sadness. Related secondary or tertiary emotions are sympathy, guilt, rejection and a sense of shame. In addition, anger was reported as well as shock (fear).

Participants feel the primary emotion of sadness when thinking of food waste caused by supermarkets and society. As Mara (23 yrs.) puts it, "I find it very bad, and I feel sadness. It is a pity that this problem can't be solved otherwise and the vegetables are thrown away." She also feels sad when she thinks of chickens coming from intensive livestock farms. The destruction of nature and the planet is mentioned by Hannah (23 yrs.) as follows, "It makes me sad because this is nature, and by putting waste out there, we destroy it rather than preserve it." In regards to the closing down of traditional small-scale food companies, Franz (60 yrs.) states: "It makes me feel sad to see a picture of the butcher shop, which was no longer profitable. It's sad in principle, because the personal relationship got lost." Additionally, the lack of appreciation towards food products are elements that make

participants such as Stefan (27 yrs.) feel sad: "As a baker's son, I feel very sad ( . . . ) in realizing that products are not seen as a craft anymore but as mass products."

Sympathy is evoked by seeing pictures of the consequences of bad weather conditions, such as loss of harvest. For example, Mara (23 yrs.) feels sympathy for farmers whose livelihoods are at risk when losing their harvest.

Participants feel guilty when they think of today's situation of livestock farming, for instance, Linda (24 yrs.), feels guilty eating meat: "I like to eat meat, and I am cautious to get good meat. However, that's not always possible. Sometimes you don't have the time, or you don't take the time," and Hannah (23 yrs.) realizes the responsibility of consumers and feels guilty that (small-scale) farmers in Austria may disappear if consumers do not consider buying from them.

Some participants reject buying products that are shipped to the supermarket, for instance, fruits and vegetables that are out of season but available all year round, as Franz (60 yrs.) states, "You get all varieties of fruits and vegetables all year round. I strongly reject that, personally." Likewise, Stefanie (54 yrs.) feels ashamed when she thinks of today's trade of food products.

Next, the primary emotion of anger was felt. For example, Linda (24 yrs.) gets angry when thinking of food waste. She admits, "Of course it makes me angry. However, I have a feeling that nothing will change because the industry regulates it."

Finally, Charlotte (28 yrs.) was shocked to learn how many desserts included the ingredient palm oil.

In addition to the emotions identified by Parrott [7], we found that most of the participants mentioned that they have doubts about the credibility of seals of quality or sustainable food brands. For instance, Stefanie (54 yrs.) states, "In the supermarket, it is not always organic or sustainable, even if it says so." Moreover, they question if imported products are still healthy and rich in nutrients, as Hannah (23 yrs.) says: "It is questionable how many vitamins are lost on their way and how the logistic chain really works."

Other negative emotions, such as annoyance, as mentioned by Charlotte (28 yrs.): "You mustn't say: Don't eat that, or buy less. It annoys people," or dissatisfaction as expressed by Mara (23 yrs.), reacting to the fact that conventional vegetables come from Spain, are found as well.

*4.3. Discussion*

The results regarding the relationship between emotions and carbon-friendly food behavior extend earlier research [27], which focused only on the emotions of guilt and pride in the context of sustainability. Our results show a great variety of evoked emotions, as identified by Parrott [7], concerning sustainable food products.

We found the positive emotions joy, optimism, pride, happiness, satisfaction, enthusiasm and desire. Additionally, affiliation, trust and feeling good were mentioned. On the other hand, the negative emotions were sadness, sympathy, guilt, rejection and a sense of shame. In addition, anger and shock as well as annoyance and dissatisfaction were mentioned.

This study has provided valuable insights into consumers' emotions regarding sustainable food products. Both positive and negative emotions were evoked; the primary emotions, joy (positive) and sadness (negative), were mentioned the most.

Nevertheless, this research is limited as only ten participants took part in the qualitative study. Therefore, an additional study on motives and emotions in combination, and the effect of emotions and motivations on behavior, is the next step. For that reason, we designed a third study assessing data on predictors (motivations, emotions) for purchase behavior of carbon-friendly food (Study 3).

## 5. Study 3

### 5.1. Method

To answer *Research Question 3* (*Can the theory of planned behavior [5], including negative and positive emotions, explain the purchase of carbon-friendly food?*), we designed a quantitative study building on the conceptual framework established previously (see Figure 1). Data were gathered through an online survey by a market research company; the questionnaire was developed based on results from Study 1 and Study 2 as well as the literature. The questionnaire was filled in by a representative Austrian sample of 1000 consumers (50.1% females, $M_{age}$ = 44.81, $SD_{age}$ = 14.55, representative for the nine Austrian countries) (Table 4). Of the 1000 Austrian consumers, most described themselves as omnivores (88.6%), eating animal as well as vegetable foods; only 11.3 percent described their diet as vegetarian (9.4%) or vegan (1.9%) (Table 4).

**Table 4.** Socio-demographics of participants in Study 3.

| Age | | M = 44.81 | SD = 14.55 |
|---|---|---|---|
| **Gender** | Female | 501 | |
| | Male | 498 | |
| | Divers | 1 | |
| **Country of Austria** | Vienna | 219 | |
| | Lower Austria | 187 | |
| | Upper Austria | 165 | |
| | Styria | 141 | |
| | Tyrol | 86 | |
| | Salzburg | 63 | |
| | Carinthia | 61 | |
| | Vorarlberg | 44 | |
| | Burgenland | 34 | |
| **Personal Net Income** | 0 EUR–500 EUR | 80 | |
| | 501 EUR–1000 EUR | 136 | |
| | 1001 EUR–1500 EUR | 191 | |
| | 1501 EUR–2000 EUR | 227 | |
| | 2001 EUR–2500 EUR | 172 | |
| | 2501 EUR–3000 EUR | 106 | |
| | More than 3000 EUR | 88 | |
| **Education** | Primary school | 49 | |
| | Vocational school | 317 | |
| | Middle school | 136 | |
| | Higher school certificate | 272 | |
| | University degree | 226 | |
| **Job Situation** | Student | 73 | |
| | Homemaker | 53 | |
| | Employed | 530 | |
| | Self-employed | 54 | |
| | Unemployed | 52 | |
| | Pensioner | 201 | |
| | Others | 37 | |
| **Persons Living in the Household** | 1 | 245 | |
| | 2 | 383 | |
| | 3 | 197 | |
| | 4 | 119 | |
| | 5 | 42 | |
| | 6 and more | 14 | |
| **Diet** | Animal and vegetable foods | 886 | |
| | Vegetarian | 94 | |
| | Vegan | 19 | |
| | Others | 1 | |
| **Grocery Shoping Primarily in Family** | | M = 5.48 | SD = 1.70 |

The measures used for data collection (Attitudes, Subjective norms, Perceived behavioral control, Emotions, Intention to purchase carbon-friendly food, Purchase of carbon-friendly food) were all multi-item constructs and answerable via a seven-point Likert scale, ranging from 1 "strongly disagree" to 7 "strongly agree." The scale Attitudes was assessed with six items (e.g., Buying carbon-friendly foods is, for me, a good idea.) following research on organic food [36]. Subjective norms were measured using four items adapted from previous research [36] (e.g., Most people I value would buy carbon-friendly food.). Perceived behavioral control was operationalized using a three-item scale, again based on items used in previous research [36] (e.g., I think it is easy for me to buy carbon-friendly food.). The scale Emotions was measured with 29 self-developed items based on the primary and secondary emotions presented in the theoretical background [7] (e.g., Purchasing carbon-friendly food, I feel joy.). For further analyses, the emotions were grouped into the six primary emotions (Sadness, Love, Anger, Joy, Surprise, Fear) that again were grouped to Negative emotions (Sadness, Anger, Fear) and Positive emotions (Love, Joy, Surprise). Intention to purchase carbon-friendly food was again measured using three items [36] (adapted, e.g., I plan to buy carbon-friendly food). The final scale, Purchase of carbon-friendly food, was assessed with eight self-developed items based on the definition of carbon-friendly food in the theoretical background [4] (e.g., I deliberately buy vegetables instead of meat to reduce $CO_2$.). Additionally to the scales above, nine Demographic variables were measured (Age, Gender, Country of Austria, Personal net income, Education, Job situation, Persons living in the household, Person primarily doing grocery shopping, Diet). Overall, reliability for all scales was very good ($0.75 < \alpha < 0.95$; see Table 5). Nevertheless, to achieve a very good Cronbach-$\alpha$, one item had to be omitted; the secondary emotion "sympathy" was excluded from the primary emotion scale Sadness because of incongruence with the remaining secondary emotions.

**Table 5.** Reliability and correlations of scales and variables.

| Scales/Variables | $\alpha$ | (1) | (2) | (3) | (4) | (5) | (6) | (7) | (8) | (9) | (10) |
|---|---|---|---|---|---|---|---|---|---|---|---|
| (1) Attitudes | 0.95 | | | | | | | | | | |
| (2) Subjective norms | 0.88 | 0.53 *** | | | | | | | | | |
| (3) Perceived behavioral control | 0.75 | 0.42 *** | 0.47 *** | | | | | | | | |
| (4) Sadness | 0.94 | −0.18 *** | −0.04 | −0.12 *** | | | | | | | |
| (5) Anger | 0.93 | −0.22 *** | −0.08 * | −0.18 *** | 0.70 *** | | | | | | |
| (6) Fear | 0.92 | −0.20 *** | −0.04 | −0.16 *** | 0.61 *** | 0.72 *** | | | | | |
| (7) Love | 0.91 | 0.26 *** | 0.42 *** | 0.31 *** | 0.17 *** | 0.17 *** | 0.18 *** | | | | |
| (8) Joy | 0.95 | 0.40 *** | 0.51 *** | 0.38 *** | −0.03 | −0.06 | −0.04 | 0.68 *** | | | |
| (9) Surprise | | 0.11 *** | 0.29 *** | 17 *** | 0.25 *** | 0.22 *** | 0.25 *** | 0.57 *** | 0.48 *** | | |
| (10) Intention | 0.93 | 0.55 *** | 0.54 *** | 0.51 *** | −0.23 *** | −0.29 *** | −0.26 *** | 0.37 *** | 0.58 *** | 0.16 *** | |
| (11) Behavior | 0.87 | 0.50 *** | 0.53 *** | 0.48 *** | −0.11 *** | −0.14 *** | −0.12 *** | 0.39 *** | 0.54 *** | 0.24 *** | 0.71 *** |

*** = $p < 0.001$, * = $p < 0.05$.

*5.2. Results*

Answering the overall research question, what factors are related to carbon-friendly consumer behavior, we tested our theoretical model by employing a structural equation model (see Figure 1). Using IBM SPSS AMOS 26 [37], an unconstrained model test was undertaken. The analysis verified the explanatory power of the theoretical model relating Attitudes, Subjective norms, Perceived behavioral control, and Negative emotions (Sadness, Anger, Fear) and Positive emotions (Love, Joy, Surprise) via Intention to purchase carbon-friendly food to Purchase behavior of carbon-friendly food (CMIN (1,1144) = 3791.42, $p < 0.001$, CMIN/df = 3.32, RMSEA = 0.05, Hoelter (0.05) = 323, CFI = 0.94). As the $chi^2$ test specified that data differed significantly from the theoretical model, additional relevant statistical tests confirmed that the significance was due to the large sample size (total 1000 respondents). For instance, CMIN/df of below 5 indicates a reasonable fit [38], and the Hoelter (0.05) measure above 200 indicates that if the sample size was reduced to 323 respondents, the $chi^2$ would not be significant [39]. Finally, the CFI, above 0.90, is a sign for an acceptable fit [40]. This confirmed that from an overall perspective, our theoretical model held (for regression coefficients in the observed model, see Table 6).

**Table 6.** Standardized regression coefficients in the observed model.

| Regressions | | | |
|---|---|---|---|
| **Predictor** | **Dependent Variable** | **β** | |
| Attitudes | Intention | 0.26 | *** |
| Subjective norms | Intention | 0.18 | *** |
| Perceived behavioral control | Intention | 0.26 | *** |
| Negative emotions | Intention | −0.25 | *** |
| Positive emotions | Intention | 0.35 | *** |
| Perceived behavioral control | Purchase behavior | 0.25 | *** |
| Intention | Purchase behavior | 0.63 | *** |

*** = $p < 0.001$.

*5.3. Discussion*

In answering Research Question 3, whether the theory of planned behavior (TPB [5]) including negative and positive emotions [7] can explain the purchase of carbon-friendly food, we find that the TPB including negative and positive emotions is an adequate theoretical vehicle to predict carbon-friendly food purchases. While motivational factors such as attitudes, subjective norms and perceived behavioral control show a rather medium positive influence on purchase intention, negative emotions also show medium negative effects and positive emotions a positive effect. Thus: (a) the more consumers associate positive evaluations with carbon-friendly food purchases; (b) the more important they perceive others to also favor carbon-friendly food purchases; (c) the more they have the feeling they actually *can* buy carbon-friendly food; (d) the more positive emotions they feel when buying carbon-friendly food; and (e) the less negative emotions experienced with carbon-friendly food purchases the higher their intention to buy carbon-friendly food and, subsequently, the higher the likelihood of their actual purchasing behavior. This is certainly in line with earlier research, in which certain emotions (e.g., the negative emotion guilt) are combined with the TPB to show their impact on specific carbon-friendly food purchases (e.g., buying organic food) [6,29]. With Study 3, we have expanded on the earlier findings. We not only used several different emotions to predict carbon-friendly food purchases, but we also focused on the whole range of carbon-friendly food purchases. This includes the well-researched buying of organic food and taking into account the length of transfer, the packaging, the production of food, and the kind of food (plant-based instead of meat).

**6. Discussion**

In light of the enormous impact food consumption has on greenhouse gas emissions, means of reducing food-related $CO_2$ need to be found to decelerate climate change. Our approach focuses on changing food consumption habits to carbon-friendly eating patterns utilizing information campaigns and legal regulation. In this respect, we investigated what antecedents and drivers determine consumers' food consumption practices. Furthermore, following the theory of planned behavior (TPB [5]) and incorporating emotions [7], we analyzed motivational aspects (attitudes, subjective norms, perceived behavioral control) and emotions by means of three empirical studies, guided by three research questions.

To answer Research Question 1 (What are the motivations to purchase carbon-friendly food?) and Research Question 2 (Which emotions emerge with the purchase of carbon-friendly food?), two qualitative studies were conducted. First, regarding the motivations, we found that ethical concerns and personal health are the main drivers for carbon-friendly food consumption. In particular, food production, for instance, of meat, and the effects of food on consumers seemed to be central. In contrast, the environmental aspect was mentioned only as a consequence of other aspects. In addition, consumers also reported negative emotions.

Therefore, the goal of Study 2 was to identify different emotions that relate to carbon-friendly food consumption. This extends previous research that focused on selected emotions with regard to sustainable consumption. Our results show that positive and

negative emotions can be evoked regarding carbon-friendly food. Using pictorial material, consumers reported the positive emotion *joy* and related emotions. They were caused by realizing the variety and quality of fresh products available, and how enjoyable producing and preparing one's own food can be. The main negative emotion discussed was *sadness*, and it was felt in relation to the consequences of industries' or consumers' behaviors on the environment. These feelings also included *guilt* or *shame*, two commonly investigated emotions. Overall, the variety of emotions and their causes revealed the importance of identifying them. For instance, the way food is produced and handled can cause positive and negative emotions. Consumer-felt control over their diet and food choice leads to positive emotions. In addition, business practices evoke consumers' emotions, which may become influential in purchase situations.

Consequently, Research Question 3 tested whether the theory of planned behavior [5], including negative and positive emotions [7], can explain the purchase of carbon-friendly food. We conducted a survey with a representative sample in Austria and found significant influences of attitudes, subjective norms and perceived behavioral control, as well as both negative and positive emotions, on the intention and subsequent purchase of carbon-friendly food. This means that consumers, in general, would purchase carbon-friendly food; they view it as something meaningful.

Regarding consequences for informational campaigns, we conclude that using emotions, preferably positive emotions, in communication with consumers will influence their intention to purchase carbon-friendly food. From a more legal and regulative perspective, we found that consumers' perceived behavioral control affects their intention and purchase behavior. Thus, the more consumers feel that they can make a difference and choose, the higher the likelihood of their carbon-friendly food purchase. Factual information, in the form of labels or packaging would help consumers learn which products are actually carbon-friendly to make their choice.

In our research, we mainly focused on negative and positive emotions. Thus, in future research, a more fine-grained inspection of different emotions, as identified in Study 2, would help understand exactly which emotions influence the purchase of particular food items. For instance, as suggested in Study 2, would *disgust* negatively influence meat purchases? In addition, it is rare that in purchase decisions only one emotion alone occurs; different emotions are felt. A further research avenue could include the analysis of mixed emotions concerning carbon-friendly food purchases.

Concluding, we can say that drivers to stir carbon-friendly food purchases are certainly motivations (attitudes, subjective norms, perceived behavioral control) as well as negative and positive emotions. It is a merit of the current research on the one hand that the theory of planned behavior [5] is extended by emotions [7] in the context of carbon-friendly food purchase and on the other hand that the whole range of carbon-friendly food purchase is included in the research model. Therefore, we can recommend necessary strategies for information campaigns and legal regulation.

**Author Contributions:** Conceptualization, E.P. and E.H.; methodology, E.P. and E.H.; formal analysis, E.P.; writing—original draft preparation, E.P. and E.H.; writing—review and editing, E.P. and E.H. All authors have read and agreed to the published version of the manuscript.

**Funding:** This research received no external funding.

**Institutional Review Board Statement:** Ethical review and approval were waived for this study, due to the fact that humans and neither animals were physically or psychologically harmed during the research. Therefore an ethical approval was not thought necessary at the institution (Vienna University of Economics and Business) where the research was undertaken.

**Informed Consent Statement:** Informed consent was obtained from all subjects involved in the study.

**Data Availability Statement:** The data presented in this study are available on request from the corresponding author. The data are not publicly available due to fact that we want to guarantee anonymity of the participants. Especially statements from the interviews could reveal participants' identity.

**Acknowledgments:** We thank Isabella Heigl and Martina Retar for their help in data collection and Erin Silangil for proofreading.

**Conflicts of Interest:** The authors declare no conflict of interest.

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
