# Peer review of "What Stirs Consumers to Purchase Carbon-Friendly Food? Investigating the Motivational and Emotional Aspects in Three Studies"

_sustainability, doi:10.3390/su13158377_

Round 1

Reviewer 1 Report

The paper deals with consumer diet and food purchase behavior, focusing on carbon-friendly food.

Introduction and theoretical framework
This section covers in high detail fundamental research on carbon-friendly food issues and addresses induced consumer behavior. However, this introduction appears to be very lengthy and needs, in the opinion of this reviewer, to be more focused on key aspects of the research hypotesis (as is done in the second part of the introduction). Therefore, overall paper readability could improve by a critical rewriting of this section aimed to reduce its length.

Materials and Methods
The section is clear and gives a correct overview of the three studies conducted. However, it is unclear to the reader if there is a methodological requirement for study 1 to include participants only with a vegetarian/vegan diet. Moreover, it should be noted that female participants account for 80% and therefore a sentence justifying such percentage should be added to avoid perception of possible methodological errors. The same goes for study 2, although in this case the percentage is less unbalanced. Study 3 methodology appears to be correct and appropriate in sample size and data processing.

Results
Results are given after each study, and seem to be clear and sound in their findings.

Discussion
This section reports the findings of the three studies to justify the overall outcome, and needs to be expanded by a critical rewriting in order to address potential further research, also because the authors state so (line 695).

Final considerations
After critical reading of the paper presented, there are some aspects that need a rearrangement or rewriting by the authors.

Reviewer 2 Report

The paper is well scientifically written, and provides various factors related to carbon-friendly consumer behavior. the following suggestion may need consider: 

(1) line 88-105, the CO2 comparison among different food or manufacturing lack of quantitative evidence. 

Reviewer 3 Report

The topic of the paper is very interesting and the empirical approaches made by the authors are very ambitious. The issues of purchase carbon-friendly food with by investigating the motivational and emotional aspects are important. It would have been more enriching recommendation can communicate for information campaigns and legislation. The author also discusses policy and managerial implications of research findings. The topic is interesting, the methods are appropriate and the author(s) findings are relevant. The literature cited in the paper is adequate and the conclusions meaningful. However, we have some comments that the authors should address before publishing.  As a reviewer, I make several suggestions for improvement. 

  • Introduction

The author(s) present a research issues based on the literature, the artcile refers to the the theory of planned behavior (TPB) theory of support the research , the main contributions, and the sequence of work.  The author(s) is suggested to improve the work appear more synthesized.  The author(s) should identify the research gap.    Please improve this.

  • Theoretical Framework

All major relevant terms are discussed in the literature, but the research hypotheses are not clearly exhibited. Overall, the paper needs to be improved with regard to the coherence of its literature review structure and organization.

The structure of the theoretical framework could be improved in several ways. Theoretical framework is suggested to consider to review research constructs and variables including attitudes, subjective norms, negative emotions, positive emotions, etc. In sake of space constraints in the publication, this section can merge with a longer theoretical framework which could have three sub-sections. Please revise this.

  • Study 1-Study3

The method selected for the empirical analysis is suitable for the study area. It is explained in a clear and understandable way. However, it should explain more specifically why the participants were selected and how the process of categorization of the variables used in the TPB research model is performed .The research methodology section including data collection and analysis is well organized. However, this research does not have research hypotheses, which are essential in any research. The way the questionnaire was carried out is not known. Please provided the questionnaire in the Appendix.

  • Study 1-Study 3 Results

This results for three studies should present the results in the logic of the research TTPB) model. Tables should be simplified by following the rules of a research. The article discusses the existence of several existing study 1-3 in the model. It is appropriate to identify, explain and discuss its significance policy and legislation implications for the study.  This study generates interesting or new findings in the area of sustainability. However, the author(s) should add the following argument for improvement:What are the major innovation / contribution of this study in either methodological or theoretical perspective comparing existing literature on the sustainability. 

  • Conclusions

The most important point focuses on the diagram of the research model. Further research applied statistics with many problem solution in the tests and how the data are interpreted (not developed this point because previous work shows enormous deficiencies that do not give credibility to the calculated data). The author(s) need to provide more meaningful information based on the findings.  This conclusions would be more meaningful if the research includes new concept and distinctive variable related to sustainability.

  • References

The references are homogenized within the text.  The reference format is suitable. The number of references is acceptable and major works that are related to the issues cited. However, the author(s) can add some important publications on the basis of my review comments.
